# First Two-Year Observational Exploratory Real Life Clinical Phenotyping, and Societal Impact Study of Parkinson’s Disease in Emiratis and Expatriate Population of United Arab Emirates 2019–2021: The EmPark Study

**DOI:** 10.3390/jpm12081300

**Published:** 2022-08-09

**Authors:** Vinod Metta, Huzaifa Ibrahim, Tom Loney, Hani T. S. Benamer, Ali Alhawai, Dananir Almuhairi, Abdulla Al Shamsi, Sneha Mohan, Kislyn Rodriguez, Judith Mohan, Margaret O’Sullivan, Neha Muralidharan, Sheikha Al Mazrooei, Khadeeja Dar Mousa, Guy Chung-Faye, Rukmini Mrudula, Cristian Falup-Pecurariu, Carmen Rodriguez Bilazquez, Maryam Matar, Rupam Borgohain, K. Ray Chaudhuri

**Affiliations:** 1Psychology & Neuroscience, Department of Neurosciences, Institute of Psychiatry, King’s College London, London, UK; 2Parkinson’s Foundation Centre of Excellence, King’s College Hospital, London, UK; 3Kings College Hospital London, Dubai, United Arab Emirates; 4Parkinson’s Association, Dubai, United Arab Emirates; 5College of Medicine, Mohammed Bin Rashid University of Medicine and Health Sciences, Dubai, United Arab Emirates; 6Higher Colleges of Technology, Dubai, United Arab Emirates; 7Stem Cell Association, Dubai, United Arab Emirates; 8Dubai Statistics Centre, Dubai, United Arab Emirates; 9People of Determination Council (POD) Council of Dubai Police, Dubai, United Arab Emirates; 10Nizams Institute of Medical Sciences, Hyderabad, Telangana, India; 11Department of Neurology, Transilvania University of Brasov, Brasov, Romania; 12Institute of Salud Carlos 111, Madrid, Spain; 13Genetic Disease Association, Dubai, United Arab Emirates

**Keywords:** young onset Parkinson’s (YOPD), Emiratis, expatriate, genetic, epigenetics, societal impact, device aided therapies, quality of life, non motor symptoms

## Abstract

Background: Phenotypic differences in Parkinson’s Disease (PD) among locals (Emiratis) and Expatriates (Expats) living in United Arab Emirates have not been described and could be important to unravel local aspects of clinical heterogenicity of PD pointing towards genetic and epigenetic variations. Objective: To investigate the range and nature of motor and nonmotor clinical presentations of PD and its impact on time to diagnosis, local service provisions, and quality of life in Emiratis and Expats in UAE, as well as address the presence of current unmet needs on relation to care and etiopathogenesis of PD related to possible genetic and epigenetic factors. Methods: a cross-sectional one point in time prospective, observational real-life study of 171 patients recruited from PD and Neurology clinics across United Arab Emirates from 2019–2021. Primary outcomes were sociodemographic data, motor and nonmotor symptoms (NMS), including cognition and sleep, and quality of life (QOL) assessments, Results: A total of 171 PD patients (52 Emiratis 119 Expats) were included with mean age (Emiratis 48.5 (13.1) Expats 64.15 (13.1)) and mean disease duration (Emiratis 4.8 (3.2) Expats 6.1 (2.9)). In the Emiratis, there was a significant mean delay in initiating treatment after diagnosis (Emiratis 1.2 (0.9) Expats 1.6 (1.1)), while from a clinical phenotyping aspect, there is a high percentage of akinesia 25 (48.1) or tremor dominant (22 (42.3)) phenotypes as opposed to mixed subtype 67 (56.3) in Expat cohorts; double tremor dominant, especially Emirati females (25%), had a predominant lower limb onset PD. Both Emirati (27.9 (24.0)) and Expat 29.4 (15.6) showed moderate NMS burden and the NMS profile is dominated by Sleep, Fatigue, Mood, Emotional well-being 3.0 (1.1) and Social Stigma 3.5 (0.9) aspects of PDQ8 SI measurements are predicted worse QOL in Emiratis, while lack of social support 2.3 (1.3) impaired QOL in Expat population. Awareness for advanced therapies was low and only 25% of Emiratis were aware of deep brain surgery (DBS), compared to 69% Expats. Only 2% of Emiratis, compared to 32% of Expats, heard of Apomorphine infusion (CSAI), and no (0%) Emiratis were aware of intrajejunal levodopa infusion (IJLI), compared to 13% of expats. Conclusion: Our pilot data suggest clinical phenotypic differences in presentation of PD in Emiratis population of UAE compared to expats. Worryingly, the data also show delayed treatment initiation, as well as widespread lack of knowledge of advanced therapies in the Emirati population.

## 1. Background

Parkinson’s disease (PD) is the second most common neurodegenerative disorder, with an increasing prevalence with age; according to a recent study by Post et al. [1], 1 in 10 people aged 45 to 100 are at risk of developing PD and 4 out of every 100 people are diagnosed with PD before the age of 50 (young onset PD) YOPD [1]. Whether or not the frequency of PD varies by race/ethnicity or gender, it is now the leading cause of disability worldwide. Unlike in European and United Kingdom Parkinson’s cohorts, Arab families have a high rate of consanguineous marriages [2] which may increase the risk of genetic phenotypes (YOPD) albeit Familial PD accounts for less than 10% of all cases of PD [3]. Some studies show low prevalence of PD in some Arab communities, especially in the Al Thuqbah region of Saudi Arabia (27 per 100,000) [4], in contrast to relatively high prevalence in north African Arabs [5] (31.4–557.4 per 100,000), whereas varied genetically heterogenous patterns were reported in Tunisian study by Gouider-Khouja N [6].

A pilot study conducted by the Movement Disorders International task force [7] identified unmet needs in the Middle East, North Africa, South Asia (MENASA) region and recommended requirement and need for multidisciplinary care, increased movement disorders specialists, educational programs, accurate epidemiologic/genetic data, awareness and availability of more advanced therapies, and suitable infrastructure to provide care to the people with PD. However, no tangible developments in relation to the aforementioned unmet needs are currently obvious, and our study aimed to address the awareness and range and nature of PD in a granular manner in a local UAE population here a comparison with settled expat communities. There are also genetic aspects that have to be considered. For instance, the LRRK2, G2019S, autosomal dominant PD with inadequate penetrance and autosomal recessive inheritance patterns were discovered in a genomic analysis of familial PD in Tunisia [8] and are now known to be prevalent in North African Arabs in Gulf cooperation council countries (GCC) with Arabic population [8,9,10]. This could be due to ancestry disparities between Arabs from the Gulf Cooperation Council and Arabs from North Africa, with the latter being considerably more closely linked to Berber ancestry [11]. As an example, Al-Mubarak et al. [11] reported no LRRK2 G2019S mutations in the Saudi population they studied.

Furthermore, epidemiologic evidence suggests that ethnicity/race may play a significant impact on genetic, epigenetic, environmental, cultural, and socioeconomic factors, which may affect the pathophysiology and symptomatic expression in PD [12,13,14]. Given its multi- ethnic population, the United Arab Emirates (UAE), particularly the Dubai area, allows our study and comparison of endophenotypic variations in carefully selected locals and expats. The results may aid in the establishment of a biobanking share initiative with the local setup, specifically to study genetic and epigenetic aspects of diseases. Aside from a few anecdotal studies, as discussed above, no obvious robust prevalence or any endophenotypic studies have been reported or been described among UAE patient cohorts to date. Ours is possibly the first UAE real-life study seeking to understand any specific clinical phenotypic (motor and nonmotor, predictors of QOL) differences in the local Emirati population compared to a wider Expat group in addition to differences in perception of treatment and delivery of care.

## 2. Methods

### Study Design

This was a cross-sectional one-point-in-time prospective, observational real-life study of 171 patients recruited from PD and Neurology clinics across United Arab Emirates from 2019 to 2021. Primary outcomes were sociodemographic data, motor and nonmotor symptoms (NMS), including cognition and sleep, and quality of life (QOL) assessments.

This study was carried out in accordance with local ethical committee guidelines. Prior to participating in the study, all patients provided written consent and all data were stored in an anonymized fashion in accordance with the ongoing UK portfolio adopted NILS longitudinal cohort study at the National Parkinson’s Centre of Excellence at Kings College Hospital in London, Dubai, in accordance with the General Data Protection Regulation (GDPR UAE). The NILS (UK) study has been authorized by local ethics committees (NRES South-East London REC3, 10,084, 10/H0808/141).

## 3. Informed Consent

Informed consent was obtained from patients/carers/all participants involved in this study.

### 3.1. Patient Selection

Patients with a confirmed diagnosis of Parkinson’s disease (PD) who met the UK PD Brain Bank criteria were recruited. Referrals to national Parkinson’s Centre of Excellence Kings College Hospital, Dubai, from all around the UAE (mainly from Dubai, Abu Dhabi Sharjah, Al Ain, Ras Al Khaimah, and others) and self-referrals were included.

Separation of Emirati and Expat groups were carried out following established local methodology. Emirates were UAE nationals and Expats were carefully selected to provide for a comparator group and only included subjects from outside Asia and settled in UAE.

### 3.2. Assessments

During the consultation, as a part of good clinical practice, standardized assessment protocols such as the demographics of (Emirati vs. Expat), age, gender, disease duration, were used, Levodopa Equivalent Daily Dose Calculation (LEDD) [15]. other scales like Hoehn and Yahr Staging (H&Y) [16], and Non-Motor Symptoms Scale (NMSS) [17], Parkinson’s Disease Questionnaire-8 (PDQ-8) [18], Kings Parkinson’s Pain Scale (KPPS) [19], PDSS (Parkinson’s Disease Sleep Scale) [20], MMSE (Mini-Mental State Examination) [21], PFS 16 (Parkinson’s Fatigue Scale) [22], and the Hospital Anxiety Depression Scale (HADS) [23] were applied. Details of these validated scales have been published elsewhere and the assessments were performed in line with the NILS assessment; a national study by the National Institute of Health Research in the UK (UKCRN No: 10,084) currently containing data for over 1600 PD patients.

## 4. Statistical Methods

Data did not fit normal distribution; thus, non-parametric statistics were applied. Descriptive statistics (frequencies and percentages, mean, standard deviation—SD-, median, and inter-quartile range—IQR) were calculated for socio-demographic and clinical variables and scale scores. Differences in scores between Emiratis and Expats were explored using Mann–Whitney and chi-square tests (significance, *p* < 0.05).

## 5. Results

In total, 171 patients of all ages and HY stages of Parkinson’s disease from across the UAE (primarily from Dubai, Abu Dhabi Sharjah, Ain, Abu Dhabi, Ras Al Khaimah, etc.) were recruited during the period 2019–2021. A total of 171 PD patients (52 Emiratis 119 Expats) were included, with mean age (Emiratis 48.5 (13.1), Expats 64.15 (13.1)) and mean disease duration (Emiratis 4.8 (3.2), Expats 6.1 (2.9)), respectively, regardless of their origin, similar to other European and Caucasian cohorts. Male preponderance (73.1%) compared to females (26.9%) is observed in both Emiratis and Expat patients, whereas disease duration in Expats cohorts 6.1 (2.9) was longer than Emiratis 4.8 (3.2) in years (Table 1).

There was a 1.5 (1.06) delay in starting PD treatment after formal diagnosis with an average delay of 1.2 (0.9) years in Emiratis and 1.6 (1.1) years in Expats, and, interestingly, we discovered at least three neurologists 3.6 (1.1) were seen by Emiratis compared to expats 2.5 (0.9) consulted after onset of symptoms, before diagnosis and initiation of PD treatment (Table 1). Surprisingly, 37% of Emirati patients were not on any treatment even after 2–5 years of diagnosis.

Emiratis appeared to have a higher rate of young onset Parkinson’s disease (PD onset below 50 years) (YOPD) and while from a clinical phenotyping aspect, there is a high percentage of akinesia 25 (48.1) or tremor dominant 22 (42.3) phenotypes as opposed to mixed subtype 67 (56.3), in Expat cohorts, double tremor dominant especially Emirati females (25%) had a predominant lower limb onset PD (Table 1).

In Table 2, the differences in the applied rating scales between Emiratis and Expats are displayed.

Based on NMSS score and staging, both Emirati (27.9) (24.0) and Expats 29.4 (15.6) had moderate NMS burden. NMS profile is dominated by Sleep, Fatigue, Mood, Emotional well-being 3.0 (1.1), and Social Stigma 3.5 (0.9) aspects of PDQ8 measurements are predicted worse QOL in Emiratis, while lack of social support 2.3 (1.3) impaired QOL in Expat population. Nocturnal pain 2.7 (1.7) dominates in Emiratis, whereas both nocturnal and radicular pain, 3.3 (1.8) and 2.3 (1.3), respectively, dominates in Expat population.

## 6. Discussion

Our study reports some key findings highlighting differences in PD presentation and delivery of Parkinson’s care among local Emirati population versus a comparator Expat PD population in UAE.

These are: Emirati PD patients tended to have young onset Parkinson’s (YOPD) 48.5 (13.1) which is lower than a global average, Khalil et al. [7]. This may underpin a genetic causation or predisposition possibly contributed to by consanguinity in Arab population, although this was not specifically studied and is certainly worthy of further larger suitable powered clinical genetic cohort studies. Moreover, 93.8% of Emiratis presented to our clinics were YOPD within 1–5 years’ duration. It is also considered that the general age of the Emirate population tends to be lower and as such there may be a bias to this observation.

The occurrence of higher proportion of Lower limb tremor (LLT) in emirate female PD is of interest. LLT has been specifically described to occur in some genetic variants of PD such as in Parkin mutation [24], as well as in those with LRRK2 [25] and the data, therefore, need more specific observation, and genetic and biomarker analysis, as well as clinical follow up of this specific cohort. There was a higher representation of nocturnal, fluctuation, and radicular pain in the LLT group. This is a preliminary finding and needs to be investigated in more granular detail. Lower back pain and shoulder pain (variants of musculoskeletal and radicular pain) have been reported in PINK1 and GBA mutation related PD cases [26]. Fluctuation is often seen at a greater level in YOPD and parkin positive cases. These factors, therefore, need exploring as Emirati patients, who were either on low levodopa doses or those who were eligible or unaware of advanced device aided therapies (DAT), respond very well either to escalating dopaminergic regime or DAT therapies.

When data on the Emirate PD are examined in a more granular fashion, it emerges that the Emirate PD, in spite of lower disease duration, have similar HY stage compared to the expat group and similar burden of overall NMS scores. Moreover, 29.0 (18.5) was seen in both Emiratis and Expats and NMS profile is dominated by Sleep, Fatigue, Mood. On the whole the overall NMS burden were similar cross both groups, and given that the emirate PD group had a significantly lower disease duration, this may mean that the clinical PD phenotype in this group may have a greater representation of the recently described NMS endophenotypes [27]. Greater understanding and clarity around this pattern of endophenotype would be important to assign sub type specific treatment and delivery of personalized medicine [28] in this group. A faster disease progression in this group, therefore, could be proposed on the basis of this observation although lower LED intake in the Emirate group could be a confounder.

Another striking feature of our study we would like to highlight is Emotional well-being and Social Stigma aspects of PDQ8 SI measurements, which were predicted worse QOL in Emiratis, while lack of social support impaired QOL in Expat population Pain in Parkinson’s is independent of disease severity so is with disease duration. Nocturnal pain Predominates in Emiratis, whereas both nocturnal and radicular pain dominate in Expat population (Figure 1).

Finally, we consider vignettes of care delivery of PD across both groups. The Emirates saw more neurologists, and, in spite of seeing at least three neurologists, there was a significant delay in initiation of treatment, even after diagnosis in general UAE PD patients (both Emiratis and Expats). Surprisingly, 33% Emiratis were not on any treatment, even after 2–5 years of diagnosis, and this observation is in conflict with the wider consensus that treatment in PD ought to be started at diagnosis as patients otherwise report progressive deterioration in QoL [29].

Delivery of care in PD is also underpinned by successful provision of advanced infusion (apo IJLI) and surgical treatments. Here, awareness of patient about these treatments is paramount and our data suggest (Figure 2) that only 25% of Emiratis are aware of the deep brain stimulation surgery (DBS), compared to 69% of Expats. Interestingly only 2% of Emiratis are aware of Apomorphine infusion treatment (CSAI), compared with 32% of Expats. Surprisingly, no (0%) Emiratis, compared to 13% of expats, were aware of intrajejunal levodopa infusion (IJLI). Out of 171 (our study sample), only 8% were treated with device aided therapies, despite the fact that nearly 50% were eligible based on Delphi 5-2-1 criteria [30]. This may be due to lack of awareness, or specialist skills or experience or advanced device aided therapy (DAT) treatment guidelines to implement these therapies. Some of the Arabic patients and care givers struggled with clinical scales/questionnaires being in English; perhaps Arabic translated ones would be beneficial.

The findings consolidate several key unmet needs related to MENASA countries as articulated in the 2020 paper by Khalil et al. [7]. In the Emirate PD, well controlled longitudinal cohort studies need to be undertaken seeking genotype phenotype correlations from a care perspective; awareness for advanced therapies needs to be improved and this needs to be a multilevel educational exercise related to both patients and health care professionals. Such access to therapies can be improved by implementation of a culturally bespoke local clinical guideline for pharmacological as well as non-pharmacological therapies for PD.

### 6.1. Why Early Diagnosis and Treatment Important in PD?

Parkinson’s disease is a progressive neurodegenerative condition attributed to progressive loss of dopaminergic neurons emerging evidence supports early intervention may help preserving the functioning of neurons helps in slowing disease progression and improving overall quality of life [31]. Early treatment depends and relies on early diagnosis; a UK autopsy study of 100 subjects who had been diagnosed with PD found a misdiagnosis rate of 24% [32], while another study [33] showed nearly 47% of PD diagnosis are incorrect when performed in primary care setting and by non-movement disorder specialists. It is necessary that the required skill set and resources are refined as early detection and treatment have potential to improve the experience and quality of life [34].

### 6.2. Clinical Benefits of Early Diagnosis and Treatment in PD?

Several studies demonstrated clinical benefit of early treatment. A multicenter controlled clinical trial of Selegiline for 24 months’ follow-up on 800 patients in 1987 demonstrated a delayed onset of disability and reduction in motor function (UPDRS) and requirement of Ldopa [35]. Early Parkinson’s disease can be managed successfully for up to five years with the use of Ropinirole alone and supplementing it with levodopa if necessary. This result is observed in a 5-year follow up study comparing the role of Ropinirole vs. L-DOPA and Benserazide [36]. In another study, Rasagiline treatment demonstrated significant improvement in motor (UPDRS) and no change in onset/frequency of adverse events in a 26-week follow-up study comparing Rasagiline vs. Placebo [37]. A 46-month SPECT study of individuals treated with Pramipexole and Carbidopa Levodopa revealed that the Pramipexole-treated group had less dopaminergic neuron degeneration than the Carbidopa-treated group, with identical UPDRS scores in both groups [38]. After a 24-month follow-up, in a PET study of patients treated with Ropinirole and Carbidopa Levodopa, the Ropinirole-treated group showed decreased dopaminergic neuron degeneration, with equivalent UPDRS ratings in both groups [39]. A 42-week follow-up study of varied multiple doses of carbidopa levodopa revealed a dose-related improvement in motor UPDRS scores [40]. With Pramipexole, there was a reduction in dyskinesia and wear-off, but the L-dopa group had a better overall score and motor score, as well as fewer side effects (freezing, somnolence, and edema) [41]. In a meta-analysis of 5247 individuals treated with dopamine agonists and levodopa, patients treated with dopamine agonists had fewer motor problems (dyskinesias or dystonia) than patients treated with levodopa [42]. Individuals treated with MAO B inhibitors had improvements in both motor scores and activities of daily living in a meta-analysis study of 3525 patients treated with MAO B inhibitors and levodopa [43]. Rasagiline 1 mg and 2 mg were compared to placebo in a 72-week follow-up study. With Rasagiline 1 mg, but not with 2 mg dosage, the early-start group had better UPDRS scores than the delayed-start group [44]. The 6.5-year extension of the TEMPO research confirmed that the early treated group had less UPDRS score degradation than the delayed onset group [45]. The intervention group experienced a slow onset of dyskinesia and had a higher frequency of dyskinesia [46]. L-dopa improves mobility and gives higher quality of life than dopamine agonists (DA) and monoamine oxidase type B inhibitors, according to a 36-month follow-up study of 1620 patients comparing levodopa and dopamine agonists and MAOB inhibitors [47] all these studies (randomized clinical trials and meta-analysis) summarized in (Table 3) supports treatment should be initiated at the time of diagnosis, delaying the treatment has worst prognostic implications (Table 3).

### 6.3. Economical Benefits of Early Diagnosis and Treatment in PD?

Early intervention is likely to have a significant impact on healthcare costs, as well as societal impact; several studies showed the impact of social healthcare burden and economic costs and quality of life is severe in the later stages of the disease, when symptoms are at their most severe, necessitating more healthcare services or caregiver support [34,35,48,49]. Motor difficulties (motor fluctuations, dyskinesias, and dystonia, which manifests as uncontrollable and sometimes painful muscular spasms) have been recognized as variables contributing to the rise in PD-related expenditures. Social, healthcare burden, and economic costs impact quality of life in patients with advanced Parkinson’s disease (APD) [48,49,50].

Patients with Parkinson’s disease (PwP) experience more unpredictable and troublesome motor and non-motor fluctuations as they progress through advanced stages, with the emergence of severe motor (progressive disability) and non-motor symptoms, such as mood, cognitive, and behavioral problems, causing a severe impact on QoL and necessitating more healthcare services or caregivers [48,49,50].

According to a study by Schrag et al. [48], the overall burden of Parkinson’s disease and healthcare resource consumption expenses grew dramatically as the disease progressed with advanced Parkinson’s disease (APD). Annual costs for early Parkinson’s disease were €2110, but for advanced Parkinson’s disease, they were about twenty times higher (€38,625), and majority of patients with advanced disease not on any device aided therapies (DAT) elderly over 70 years old [48]. A Spanish study by Zecchinelli et al. [51] revealed roughly 30% of Parkinson’s patients are in advanced stages, and the cost of illness rose sharply, primarily due to costs linked with in-patient treatment and nursing homes because advanced-stage patients are bedridden, wheelchair-bound, or hospitalized [52,53]. The primary drivers and determinants of the socio-economic burden of PD were hospitalization, nursing care, drug costs, indirect costs (loss of work, etc.), predictors of quality of life, societal socio-economic impact healthcare burden, and QOL in PwP [54,55,56] (Figure 3).

A study by Popov et al. [57] looked at costs of PD illness and societal burden in a cohort of 100 patients showed over all annual burden of 1 billion euros with direct costs accounting to 67% and indirect costs accounting 33% and main drivers of the burden being informal care and drugs [58]. Another UK study by McCrone [59] et al. showed the informal care compared to formal (80% vs. 20%) impact on societal burden and the main predictors being male gender, level of disability and non-motor symptoms like depression [59], as well as adherence to oral medications, especially in elderly patients with advanced disease where they have to take several pills multiple times

Strong predictors of socio-economic burden 61% of PD patients were non-adherent to oral therapy and medical costs were significantly higher among non-adherent versus adherent ($15,826 vs. $9228) [60]. A multicenter (France, Germany, and UK) observational study by Pechevis et al. [61] showed dyskinesia (motor complications measured using UPDRS scale) was associated with significant socio-economic and societal burden and increasing total healthcare costs with each unit increase in dyskinesia score led to 562 euros additional costs per patient over a 6-month period [61].

The economic and clinical evidence gathered in the literature shows and confirms that early diagnosis and initiation of treatment is crucial, halts risk of disease progression, and reduces the effects on QOL. This can potentially reduce treatment costs if possible non-oral therapeutic device aided therapies are offered to patients as they progress to an advanced stage before significant deterioration has occurred. Patients’ QOL and well-being are improved when the Multidisciplinary care approach and timely referrals to a movement disorders specialist with expertise in PD, as selection of patients for advanced device aided therapies (IJLI, CSAI, DBS) are likely to be most effective and patients are likely to be more complaint with these therapies.

## 7. Conclusions

Our study highlights heterogenetic and endophenotype variations of Parkinson’s disease in UAE population comparing local Emirati and Expat populations. Our study identifies the importance of early diagnosis, prompt treatment initiation, which has huge societal socio-economic impact, and healthcare burden. Moreover, timely implementation of advanced therapies help delay PD disease progression. A bio banking share initiative with the local setup specifically to study genetic and epigenetic aspects focusing on: GBA, LRRK2, Parkin gene mutation. Screening of Emirati patients with young onset Lower limb tremor dependent Parkinson’s disease would be beneficial, identifying these endophenotypes is paramount as these patients will respond very well to dopaminergic dose escalation or to advanced device aided therapies and also helps to formulate gene-targeted therapies. Setting up a local expert committee panel, implementation of national treatment protocols involving patients and care giver groups (expert patient panel) will help empower patients and caregivers.

## Figures and Tables

**Figure 1 jpm-12-01300-f001:**
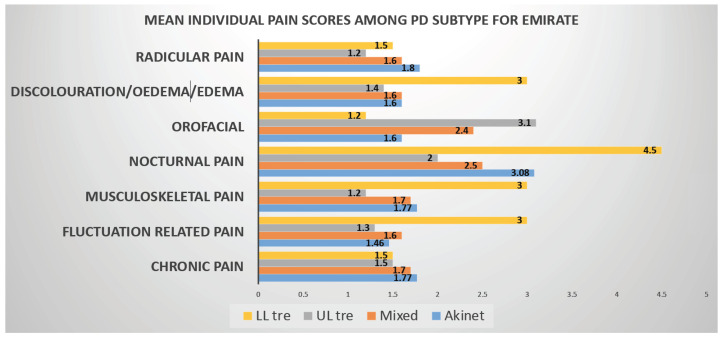
Graphical representation of PD subtype vs. pain domains (KPPS) in Emiratis. In this graph it can be seen that Nocturnal pain (NP) is found to be higher in all the PD Subtypes where as Radicular pain scores high in Tremor Dominant Subtype.

**Figure 2 jpm-12-01300-f002:**
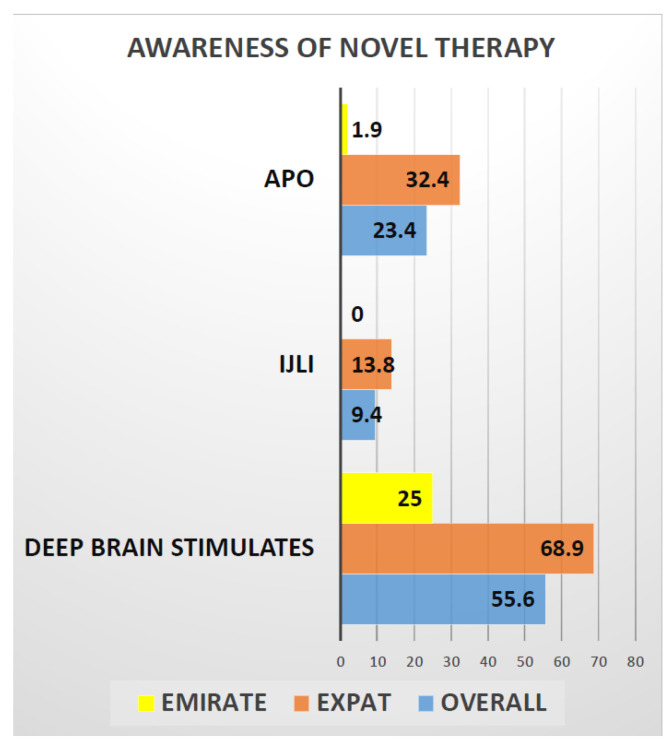
Graphical representation of the awareness of novel therapies in PD treatment among the Emiratis and Expats. This graph shows only 25% of Emirates are aware of deep brain stimulation test compared to 69% Expats and 2% of Emirates are aware of Apomorphine pump treatment compared to 32% of Expats. Whereas Not even a single ( 0%) Emirati is aware of Ileo-jejunal levodopa infusion compared to 13% of expats.

**Figure 3 jpm-12-01300-f003:**
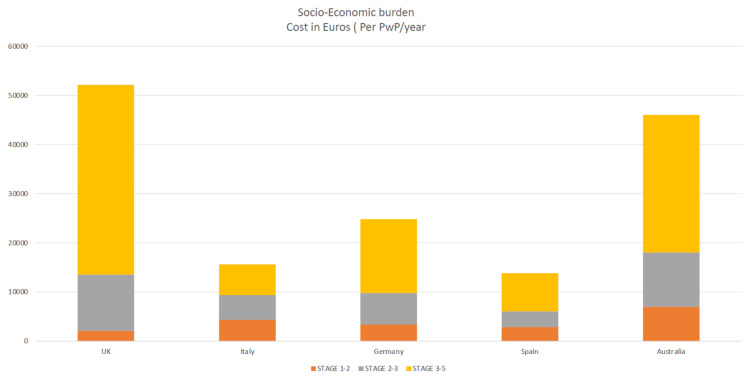
highlighting socio-economic burden increases as disease progress (yellow).

**Table 1 jpm-12-01300-t001:** Demographic and clinical variables by origin of the patients.

Age	59.4 (14.9)	48.5 (13.1)	64.15 (13.1)	<0.001 *
Disease Duration (years)	5.7 (3.05)	4.8 (3.2)	6.1 (2.9)	<0.001 *
H&Y	2.5 (0.5) ^a^	2.5 (0.5) ^a^	3 (0.5) ^a^	<0.001 **
LED	752.1 (457.8)	473.08 (473.7)	874.03 (394.06)	<0.001 *
Delay in treatment (years)	1.5 (1.06)	1.2 (0.9)	1.6 (1.1)	0.03
Number of neurologists seen	2.9 (1.1)	3.6 (1.1)	2.5 (0.9)	<0.001 *
PD subtypes				<0.001 **
Akinetic	51 (29.8) ^b^	25 (48.1) ^b^	26 (21.8)	
Tremor	48 (28.1) ^b^	22 (42.3) ^b^	26 (21.8)	
Mixed	72 (42.1) ^b^	5 (9.6) ^b^	67 (56.3)	

All values are expressed as mean (standard deviation), except ^a^ median (inter-quartile range) and ^b^ frequency (percentage). * Mann-Whitney test; ** chi-square test.

**Table 2 jpm-12-01300-t002:** Scales scores and differences by origin of the patients.

Variables	Total Sample (N = 171)	Emiratis (N = 52)	Expats (N = 119)	*p* *
NMSS Total	29.0 (18.5)	27.9 (24.0)	29.4 (15.6)	0.030
Cardiovascular	2.0 (1.5)	1.4 (1.5)	2.3 (1.5)	<0.001
Sleep/fatigue	4.0 (2.7)	5.2 (3.6)	3.5 (2.1)	0.006
Mood/Apathy	2.7 (3.0)	3.2 (4.9)	2.4 (1.6)	0.028
Perceptual problems/ Hallucinations	2.1 (1.6)	1.5 (1.4)	2.4 (1.6)	<0.001
Attention/memory	2.4 (2.4)	1.6 (1.8)	2.7 (2.6)	<0.001
Gastrointestinal	3.0 (2.0)	2.6 (2.4)	3.1 (1.7)	0.013
Urinary	3.0 (2.0)	2.2 (3.4)	3.3 (4.6)	<0.001
Sexual function	7.5 (5.4)	8.0 (6.5)	7.2 (4.8)	0.954
Miscellaneous	2.2 (1.8)	2.0 (2.3)	2.3 (1.5)	0.012
PDQ-8 SI	61.6 (21.2)	54.6 (18.3)	64.6 (21.6)	0.002
Bodily Discomfort	2.4 (1.2)	2.3 (1.2)	2.4 (1.2)	0.645
Communication	2.0 (1.3)	1.6 (1.2)	2.1 (1.3)	0.038
Cognition	2.0 (1.3)	1.4 (1.0)	2.2 (1.3)	<0.001
Social Support	2.0 (1.3)	1.5 (1.2)	2.3 (1.3)	<0.001
Stigma	3.4 (1.0)	3.5 (0.9)	3.3 (1.1)	0.266
Emotional well-being	3.2 (1.1)	3.0 (1.1)	3.1 (1.1)	0.625
Activities of daily living	2.5 (1.3)	2.0 (1.1)	2.7 (1.3)	0.003
Mobility	2.2 (1.3)	1.7 (1.2)	2.4 (1.3)	0.007
KPPS Total	16.1 (8.0)	13.3 (7.2)	17.3 (8.1)	0.002
Musculoskeletal pain	2.1 (1.2)	1.7 (1.1)	2.2 (1.2)	0.002
Chronic pain	2.1 (1.2)	1.6 (0.9)	2.3 (1.2)	<0.001
Fluctuation-related pain	2.1 (1.2)	1.6 (0.8)	2.3 (1.2)	<0.001
Nocturnal pain	3.1 (1.8)	2.7 (1.7)	3.3 (1.8)	0.018
Orofacial pain	2.4 (2.4)	2.3 (3.9)	2.4 (1.4)	<0.001
Discoloration/oedema	2.2 (1.3)	1.7 (1.0)	2.4 (1.5)	0.002
Radicular Pain	2.1 (1.3)	1.6 (1.0)	2.3 (1.3)	<0.001
PFS-16	10.5 (2.8)	11.1 (2.7)	10.2 (2.8)	0.089
MMSE	28.1 (2.9)	29.2 (2.1)	27.6 (3.0)	<0.001
PDSS	70.7 (19.6)	77.3 (17.2)	67.8 (19.9)	0.003
HADS-Anxiety	9.4 (2.2)	9.9 (2.2)	9.2 (2.2)	0.079
HADS-Depression	8.2 (2.8)	7.5 (2.8)	8.5 (2.7)	0.023

* Mann-Whitney test.

**Table 3 jpm-12-01300-t003:** Studies showing clinical benefits of early diagnosis of Parkinson’s disease.

Year	Study	Outcome
1993	DATATOP	Selegiline compared with placebo, 24-month follow up of 800 patients showed Selegiline delayed the onset of disability and reduction in motor (UPDRS) and requirement of L-dopa
2000	RASCOL et al.	A 5-year follow up study comparing Ropinirole vs. L-dopa and Benserazide, patients treated with Ropinirole had longer time to dyskinesia’s and no significant difference or change in motor scores or quality of between two groups
2002	TEMPO	A 26-week follow up study comparing Rasagiline vs. Placebo, Rasagiline treated group showed significant improvement in motor (UPDRS) and no difference in onset/frequency of adverse events.
2002	CALM-PD-CIT	A 46-month follow up SPECT study of patients treated with Pramipexole and carbidopa Levodopa showed less dopaminergic neuron degeneration in Pramipexole treated group with similar UPDRS scores in both groups.
2003	REAL-PET	A 24-month follow-up PET study of patients treated with Ropinirole and Carbidopa Levodopa showed less dopaminergic neuron degeneration in Pramipexole treated group with similar UPDRS scores in both groups.
2004	ELLDOPA	A 42-week follow up study of various multiple doses of carbidopa levodopa showed significant improvement in motor UPDRS scores in a dose related fashion.
2008	STOOWE et al.	A meta-analysis study 5247 patients treated with dopamine agonists and Levodopa; patients treated with dopamine agonists has less motor complications (dyskinesia’s, dystonia)
2008	Ives et al.	A meta-analysis study 3525 patients treated with MAO B inhibitors and Levodopa; patients treated with MAO B inhibitors have improvements in both motor scores and activities of daily living.
2009	ADAGIO	A 72-week follow up study comparing Rasagiline 1 mg and 2 mg compared with placebo showed Improved UPDRS scores in the early-start group compared to delayed-start group, with Rasagiline 1 mg but not with 2 mg dosage
2009	Hauser et al.	A 6.5-year extension of TEMPO study indeed showed early treated group has less worsening of UPDRS scores compared to delayed onset
2014	PD MED TRIAL	A 36-month follow up study of 1620 patients comparing levodopa and dopamine agonists and MAOB inhibitors showed L-dopa improves mobility and provides better quality of life compared to dopamine agonists (DA) and monoamine oxidase type B inhibitors (MAOBI)

## Data Availability

Raw data were generated at Kings College hospital London, Dubai. Derived data supporting the findings of this study are available from the corresponding author VM on request.

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
