# Peer review of "First Two-Year Observational Exploratory Real Life Clinical Phenotyping, and Societal Impact Study of Parkinson’s Disease in Emiratis and Expatriate Population of United Arab Emirates 2019–2021: The EmPark Study"

_jpm, 2022, doi:10.3390/jpm12081300_

Round 1
Reviewer 1 Report
This article presents the clinical phenotype, diagnosis, treatment, and quality of life of PD in UAE population comparing local Emirati and Expat populations. Interestingly, the differences reflected the high prevalence of juvenile (familial) Parkinson's disease in the Arab region, including Emiratis. The strengths of this manuscript are that it is well written, clearly focused and that the flow of information can easily be followed.
1. In juvenile or hereditary PD, PARK2 (parkin) and PARK8 (LRRK2) patients respond better to L-dopa, so it is possible that there are Emirati PD patients with low L-dopa doses or low awareness of advanced therapy. It would be good to have a mention in the DISCUSSION of the clinical features of juvenile PD (comparison with typical, and sporadic PD, and responsiveness to L-dopa).
2. The second and third paragraphs in the Result are duplicates.
3. P6 line 25: I think "Pramipexole-treated group" should be "Ropinirole-treated group".
Reviewer 2 Report
The authors in manuscript entitled “First two-year observational exploratory real life clinical pheno-2 typing, and societal impact study of Parkinson’s Disease in 3 Emiratis and Expatriate population of United Arab Emirates 4 2019-2021: the Em-Park Study” have studied that data from the Emirati peoples provide better care to the people with PD.
Strengths of the study:
- This manuscript has written in descriptive manner.
- The reference list is updated, but there is need to add some more references.
- The research article described that this study highlights heterogenetic, endophenotype variations of Parkinson’s disease in UAE population comparing local Emirati and Expat populations that conducted by the Movement Disorders International task force and recommended requirement and need for multidisciplinary care, increased movement disorders specialists, educational programmes, accurate epidemiologic/genetic data, aware-ness and availability of more advanced therapies, availability of superior health care re-sources and infrastructure, and higher levels of awareness to provide care to the people with PD. Results show delayed treatment initiation and lack of knowledge of advanced therapies in the Emirati population.
There are some issues with this article, if these issues are going to resolve then the quality of the paper is suitable for publication.
1) In a part of the introduction, it should be crisp and brief about the focused study. It has included more unnecessary details.
2) Some recent references should be included.
3) There are few typos and English and grammar errors which should be rectify.
4) Figure quality must be improved, Figure is blur and not as per quality of the Journal.
5) Conclusion prospective must be written crisp and clear.
